# Consistency Guided Knowledge Retrieval and Denoising in LLMs for Zero-shot Document-level Relation Triplet Extraction

Submission Id: 2194

## ABSTRACT

Document-level Relation Triplet Extraction (DocRTE) is a fundamental task in information systems that aims to simultaneously extract entities with semantic relations from a document. Existing methods heavily rely on a substantial amount of fully labeled data. However, collecting and annotating data for newly emerging relations is time-consuming and labor-intensive. Recent advanced Large Language Models (LLMs), such as ChatGPT and LLaMA, exhibit impressive long-text generation capabilities, inspiring us to explore an alternative approach for obtaining auto-labeled documents with new relations. In this paper, we propose a Zero-shot Document-level Relation Triplet Extraction (ZeroDocRTE) framework, which **Gen**erates labeled data by **R**etrieval and **D**enoising **K**nowledge from LLMs, called GenRDK. Specifically, we propose a chain-of-retrieval prompt to guide ChatGPT to generate labeled long-text data step by step. To improve the quality of synthetic data, we propose a denoising strategy based on the consistency of cross-document knowledge. Leveraging our denoised synthetic data, we proceed to fine-tune the LLaMA2-13B-Chat for extracting document-level relation triplets. We perform experiments for both zero-shot document-level relation and triplet extraction on two public datasets. The experimental results illustrate that our GenRDK framework outperforms strong baselines. The code and synthetic dataset will be released on GitHub.

## CCS CONCEPTS

• **Information systems → Information retrieval**.

## KEYWORDS

Document-level Relation Triplet Extraction, Zero-shot Learning, Knowledge Denoising, Large Language Models, Synthetic Data

## 1 INTRODUCTION

Relation Triplet Extraction (RTE) aims to extract the entity pair and the semantic relation type from the unstructured text, which plays a vital role in various downstream Natural Language Processing (NLP) applications, including knowledge graph construction and information retrieval [15, 22, 29]. Although previous approaches achieve reasonable performance [25, 34, 37], they heavily rely on the large-scale human-annotated corpus, which is inevitably time-consuming and labor-intensive. Therefore, recent efforts tend to focus on the Zero-shot Relation Extraction (ZeroRE) [1, 21, 36] and Relation Triplet Extraction (ZeroRTE) [2] tasks.

In the zero-shot scenario, the model needs to generalize to unseen relation types in the absence of available human-annotated training data. To solve this challenge, most of the existing methods attempt to reformulate the ZeroRE task to other tasks, such as the reading comprehension [13], textual entailment [19], and close question answering [9] tasks. Although these approaches show promising

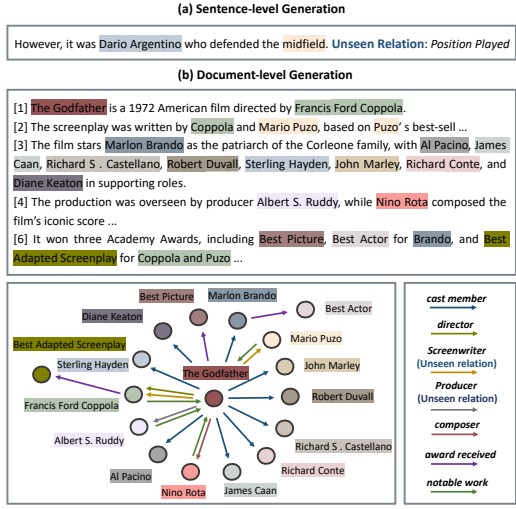

**(a) Sentence-level Generation**

However, it was Dario Argentino who defended the midfield. **Unseen Relation:** *Position Played*

**(b) Document-level Generation**

[1] The Godfather is a 1972 American film directed by Francis Ford Coppola.
[2] The screenplay was written by Coppola and Mario Puzo, based on Puzo's best-sell ...
[3] The film stars Marlon Brando as the patriarch of the Corleone family, with Al Pacino, James Caan, Richard S . Castellano, Robert Duvall, Sterling Hayden, John Marley, Richard Conte, and Diane Keaton in supporting roles.
[4] The production was overseen by producer Albert S. Ruddy, while Nino Rota composed the film's iconic score ...
[6] It won three Academy Awards, including Best Picture, Best Actor for Brando, and Best Adapted Screenplay for Coppola and Puzo ...

**Figure 1: Comparison of sentence-level [2] and document-level data generated. In sentence-level synthetic data, there exists merely one relation triplet within a sentence. In the case of document-level synthetic data, there are more than 22 relation triplets distributed across different sentences. Entities and relations are marked in different colors.**

performance, they make the unrealistic assumption that the entity pairs are readily accessible. Hence, existing endeavors [2] seek to explore the ZeroRTE task by generating synthetic data based on descriptions of previously unseen relation types.

However, the methods mentioned above primarily concentrate on sentence-level ZeroRE and ZeroRTE tasks, assuming that the entities and relations are confined within a single sentence. In practice, numerous valuable relational facts are expressed across multiple sentences, which cannot be extracted using the aforementioned zero-shot approaches. Therefore, we introduce a Zero-shot Document-level Relation Triplet Extraction task (ZeroDocRTE), which aims to extract relation triplets with unseen relation types in a whole document, formed as: (head entity, tail entity, and unseen relation type). In contrast to sentence-level ZeroRTE, ZeroDocRTE is more challenging due to the intricate semantic contexts and discourse structures of the document. Inspired by the impressive long-text generation capabilities of recent advanced Large language models (LLMs), such as ChatGPT and LLaMA, we leverage existing LLMs to obtain auto-labeled documents with new relations. Different from sentence-level synthetic data generation [2], document-level synthetic data need to contain relation triplets spanning multiple sentences, which can be seen in Figure 1.

To address this task, we propose a ZeroDocRTE framework, which **Gen**erates labeled data by **R**etrieval and **D**enoising **K**nowledge

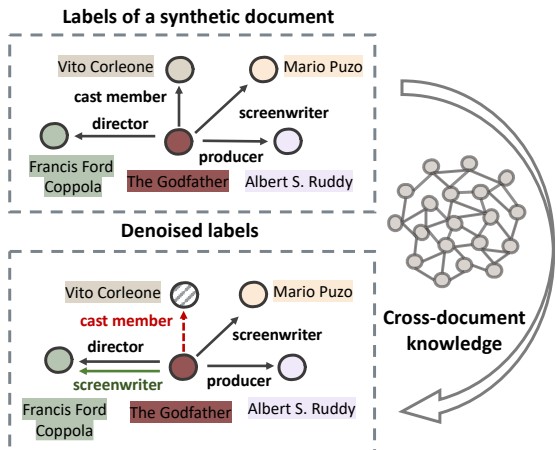

**Figure 2: The original and denoised labels of a synthetic sample. Two main types of noise are reduced by our consistency-guided cross-document knowledge denoising strategy. One is reducing the incorrect triplet as shown in the red dotted line** *(The Godfather, Vito Corleone, cast member),* **and another is adding the missing triplet as shown in the green solid line** *(The Godfather, Francis Ford Coppola, screenwriter).*

from LLMs, called GenRDK. Specifically, we propose a chain-of-retrieval prompt to guide ChatGPT to generate labeled long-text data step by step. While we can automatically generate a wide range of synthetic data, the process inevitably introduces noisy labels. As seen in Figure 2, there are many incorrect relational facts in synthetic data due to the hallucination problem [5] of LLMs. Therefore, to mitigate false labels of synthetic data, we propose a consistency-guided cross-document knowledge denoising strategy. First, a pre-denoising DocRTE model is trained with seen relation data to obtain pseudo labels of synthetic data. Next, we construct cross-document knowledge graphs according to the pseudo labels and original labels of synthetic data. By observing that the same relational fact can be expressed in different forms across different synthetic documents, we calculate consistency scores to evaluate the reliability of relational facts. Last, we prune unreliable relational facts and relabel the synthetic data. As seen in Figure 2, the missing relation triplet can be added by cross-document knowledge, and the incorrect relation triplet can be reduced by consistency scores. We proceed to fine-tune the LLaMA2-13B-Chat by our denoised synthetic data for extracting document-level relation triplet.

The main contributions of our work are summarized as follows:

- We explore a challenging Zero-shot Document-level Relation Triplet Extraction (ZeroDocRTE) task and propose a novel framework that generates synthetic data by retrieving and denoising the implicit knowledge from LLMs.
- We propose a chain-of-retrieval prompt for guiding ChatGPT to generate documents that contain intricate semantic contexts and various relation triplets step by step.
- We propose a consistency-guided cross-document knowledge denoising strategy aimed at enhancing the quality of synthetic data through the reduction of unreliable relational facts and the addition of missing relational facts.

- We perform our framework on zero-shot document-level relation and triplet extraction tasks. The experimental results illustrate that our GenRDKachieves significant performance improvements over competitive baselines.

## 2 RELATED WORK

**Sentence-level Relation Triplet Extraction.** Sentence-level RTE aims to extract the entities and relations from a single sentence simultaneously. Conventional works mainly focus on supervised relation triplet extraction[25, 34, 37]. Although these models achieve great success in the sentence-level RTE task, they heavily rely on the large-scale corpus that needs cumbersome data cleaning and time-consuming labeling. Moreover, in realistic scenarios, there might be relation types that do not have training data, yet are shown in the inference process, called unseen relation types. To solve this issue, recent research efforts have sought the Zero-shot Relation Extraction (ZeroRE) task that aims to classify the unseen relation type between the given entity pair in a sentence [1, 13, 19, 21, 36]. Nevertheless, these approaches assume the availability of ground-truth head and tail entities within a sentence, which is not always satisfied in the application. Thus, scholars [2] first propose the zero-shot setting for the RTE by using synthetic examples. However, the aforementioned techniques primarily concentrate on ZeroRE and ZeroRTE tasks at the sentence level, posing challenges for their direct application to zero-shot document-level relation and triplet extraction tasks.

**Document-level Relation Extraction.** Existing approaches mainly focus on the Document-level Relation Extraction (DocRE) task, which employs the transformer-based [12, 14, 31, 38] and the graph-based [3, 7, 18, 20, 24, 27, 28, 32, 33] models to extract contextual and non-local structural information for aggregating entity representations [12, 14, 31, 38]. While these models have achieved remarkable success in the task of DocRE, they necessitate prior knowledge in the form of ground-truth entity positions. Then, recent works attempt to extract entities and relations jointly in an end-to-end manner [8, 35]. However, the aforementioned methods depend on extensive supervised data and do not apply to ZeroDocRTE and ZeroDocRE tasks. To solve these challenging settings, we propose a novel framework, which synthesizes documents and labels by retrieving the latent knowledge of ChatGPT. To mitigate the issue of noise during the generation process, we introduce a consistency-guided knowledge denoising strategy, which can further improve the quality of synthetic data.

## 3 METHODOLOGY

In this section, we introduce our proposed framework in detail. As shown in Figure 3, our GenRDKcontains four key steps: (1) Chain-of-retrieval prompt to generate labeled data; (2) Training a pre-denoising model to obtain pseudo labels; (3) Consistency-guided cross-document knowledge denoising; (4) Training the relation triplet extractor.

### 3.1 Problem Formulation

Given a dataset $D = D_s \cup D_u$ with a set of pre-defined relation types $R = R_s \cup R_u, R_s \cap R_u = \emptyset$. $D_s$ is a seen dataset with only seen relation type sets $R_s$, $D_u$ is an unseen dataset with both seen

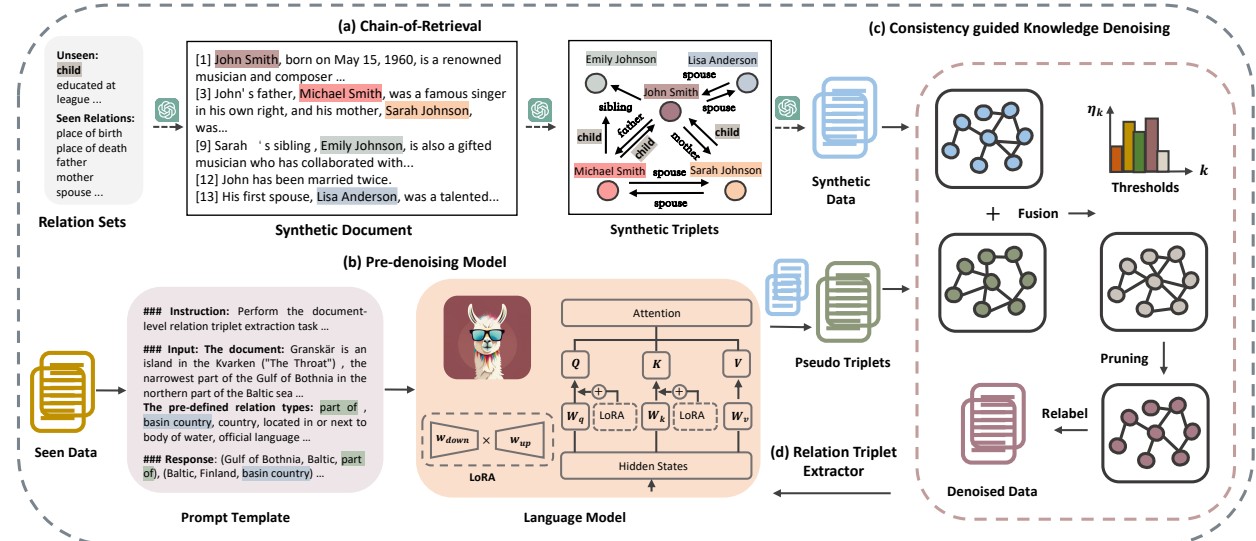

**Figure 3: The overview of our GenRDK framework. It contains four key parts as follows: (a) Chain-of-retrieval prompt for guiding ChatGPT to generate labeled data step by step; (b) Training the pre-denoising model based on LLaMA2-13B-Chat with LoRA; (c) Consistency-guided cross-document knowledge denoising strategy. (d) Training the relation triplet extractor with the denoised synthetic data.**

$R_s$ and unseen relation type sets $R_u$. Given a document $d_i \in D_u$, the zero-shot document-level relation triplet extraction aims to extract relation triplets with unseen relation types, formulated as $\{(e_s, e_o, r_k)|e_s, e_o \in E_i, r_k \in R_u\}$, where $R_u$ is the set of unseen relation types, $e_s$ is the head entity, $e_o$ is the tail entity, $E_i$ is the set of entities of document $d_i$.

## 3.2 Chain-of-Retrieval Prompt

Large Language Models (LLMs) have shown powerful zero-shot generalization ability in various NLP applications, which benefit from large-scale pre-training. Recent approaches [2, 6] exploit the implicit knowledge of LLMs to generate the synthetic data for the downstream tasks, formulated as follows:

$$[s_i, y_i] = LLM(q_i), \quad (1)$$

where $q_i$ is the query input sequence, $s_i$ and $y_i$ is the sentence and label generated by the large language model.

These methods mainly focus on generating sentence-level data that usually have a single semantic structure[2, 6]. However, synthetic data for document-level relation triplet extraction usually contain complex semantic structures and various relation triplets. Therefore, we propose a chain-of-retrieval prompt that partitions the complex generation problem into a sequence of simple questions, which can be seen in Figure 4. The process of generating synthetic data is as follows:

- For each unseen relation type $r_i \in R_u$, we prompt ChatGPT to select several relations $\{r_{ij}\}_{j=1}^{n_i}$ that most related to the unseen relation type $r_i$ from the relation set $R$, where $n_i$ is the number of selected relations.
- We guide ChatGPT to generate a fictional document $d_{ik}$ that contains the unseen relation type $r_i$ and related relations

$\{r_{ij}\}_{j=1}^{n_i}$, where $k$ is the index of document for unseen relation type $r_i$. To enhance the diversity of the generated document, we set the hyper-parameter *temperature* of ChatGPT to 1 in this step.

- Corresponding to the generated document $d_{ik}$, we prompt ChatGPT to extract the entity set $E_k$ with the pre-defined entity types.
- We prompt ChatGPT to extract all types of relation triplets $\{(e_s, e_o, r_l)|e_s, e_o \in E_k, r_l \in R\}$ based on the above document $d_k$ and entity set $E_k$.
- After obtaining relation triplets and documents, we prompt ChatGPT to present the reasoning explanation of each relation triplet, formulated as $(e_s, e_o, r_l, a_c)$.
- We then prompt ChatGPT to present the support sentences shown in the generated document $d_i$, which can be formulated as $(e_s, e_o, r_l, h_p)$.
- We guide ChatGPT to generate the final structured labels based on all the above information.

## 3.3 Pre-denoising Model

Despite the ChatGPT [1] can generate promising synthetic data, it can also produce plausible yet incorrect factual information, which is called hallucination in LLMs [5]. Thus, to further improve the quality of synthetic data, we train a pre-denoising model by data with seen relations to generate pseudo labels.

As shown in Figure 3 (b), we leverage the seen dataset $D_s$ to fine-tune the LLaMA2-13B-Chat [2] with Low-Rank Adaptation (LoRA) [11], which approximates the weight update by inserting trainable low-rank matrics into transformer layers [10]. During the fine-tuning process, we introduce a random combination strategy to

---
[1] https://chat.openai.com/
[2] https://ai.meta.com/llama/

| Step | Prompt Template | Output of ChatGPT |
|---|---|---|
| 1 | Select relation types that most **related** to the relation type <Unseen Relation Type> from the following relation sets <All Relation Types>. | <Related Relations>:
1. **director**: Screenwriters often collaborate closely with directors in the filmmaking process, as directors bring the script to life on the screen. This relationship is essential in the film industry.
2. **producer:** Producers work closely with screenwriters to develop and ...
3. **cast member:** Screenwriters may have interactions ...
... |
|  | <Unseen Relation Type>: screenwriter |  |
|  | <All Relation Types>: head of government, country, place of birth, place of death, father, mother, spouse, country of citizenship, continent, instance of, ... |  |
| 2 | Generate one fictional Wikipedia style **paragraph** that contains at least 6 sentences and describes one or more following relation types: <Unseen Relation Type> and <Related Relations>. Provide them in JSON format with just following keys: title, context. | <Synthetic Document>:
{"**title**": "The Godfather",
"**context**": "**The Godfather** is a 1972 American film directed by **Francis Ford Coppola**. The screenplay was written by **Coppola** and **Mario Puzo**, based on Puzo' s best-selling novel of the same name. The film stars **Marlon Brando** as the patriarch of the .... "} |
| 3 | Extract the **entities** in your above generated document. Provide them in List of JSON format with the following keys: entity, entity type. The entity type can be one of the following types: <Entity Types>. | <Entities>:
[{"**entity**": "The Godfather" "**entity type**": "Miscellaneous"},
{"**entity**": "Francis Ford Coppola" "**entity type**": "Person"},
{"**entity**": "Mario Puzo" "**entity type**": "Person"},
{"**entity**": "Marlon Brando" "**entity type**": "Person"},
{"**entity**": "Al Pacino" "**entity type**": "Person"},
{"**entity**": "James Caan" "**entity type**": "Person"}, ... ]} |
|  | <Entity Types>: Organization, Location, Time, Person, Miscellaneous, Number Blank. |  |
| 4 | Present the **relation triplets** as: (head entity, tail entity, relation type). The relation type can be one or more of following relation types: <All Relation Types>. | <Triplets>:
1. (The Godfather, Francis Ford Coppola, **director**)
2. (The Godfather, Mario Puzo, screenwriter)
3. (The Godfather, Marlon Brando, **cast member**) ... |
| 5 | According to the above information, present the reasoning **explanation** of each relational triplet. | <Reasons>:
1. (The Godfather, Mario Puzo, screenwriter)
    **Explanation**: Mario Puzo and Francis Ford Coppola wrote ... |
| 6 | Present **support sentence** index for each extracted relation triple that shown in the generated document. | <Support Sentence>:
1. (The Godfather, Mario Puzo, screenwriter)
    **Support Sentence Index**: Sentence **2** ... |
| 7 | Organize the above triplet information in the List of JSON format with the following keys: head entity, tail entity, relation type, reasoning explanation of each relation triplet, index of supporting sentence that shown in document. | <Synthetic Labels>:
[{"**head entity**": "The Godfather", "**tail entity**": "Mario Puzo",
"**relation type**": "screenwriter", "**reasoning explanation**": "Mario Puzo was another co-writer of the screenplay for ...", "**index of supporting sentence**": 2 }, ... ]} |

**Figure 4: A sample of the proposed chain-of-retrieval. The generation procedure is a chatting process, which means each step contains the memory of the previous steps.**

dynamically compose the relation set. In this way, we can enhance the diversity of training data. Specifically, we partition the seen relation set $R_s$ into multiple relation groups. This partition can be expressed as:

$$R_s = [R_1, R_2, ..., R_j, ..., R_m], \quad (2)$$

where $m$ is the number of relation groups. We take each relation group $R_j = \{r_{ik}\}_{k=1}^{z}$ as the input along with the document content. The fine-tuning process of each sample can be expressed as:

$$\hat{M} \leftarrow Train(M, I, d_i^s, R_j, T_{ij}^s), \quad (3)$$

where $M$ denotes the backbone model, $I$ is the task description of DocRTE task, $d_i^s$ is the $i$-th document in seen relation dataset $D_s$, $R_j$ is the $j$-th relation group, $T_{ij}$ represents the relation triplets of $j$-th relation group in the $i$-th document, $\hat{M}$ is the fine-tuned model. To obtain the pseudo labels, we perform inference on synthetic data with our pre-denoising model, formulated as follows:

$$P_i = \hat{M}(I, d_i^u, R_u), \quad (4)$$

where $\hat{M}$ is the pre-denoising model, $d_i^u$ is the $i$-th document in unseen dataset $D_u$, $R_u$ is the unseen relation set, $P_i$ is the pseudo labels of the document $d_i$.

## 3.4 Consistency guided Knowledge Denoising

We observe that different documents in synthetic data might be generated by the same relation fact, as shown in Figure 5. Inspired by this phenomenon, we attempt to supplement the losing positive relational fact in a single document with cross-document knowledge. Therefore, we propose a consistency-guided cross-document knowledge denoising strategy.

We aim to construct two knowledge graphs $KG_s$ and $KG_p$ according to the relational facts in pseudo labels and synthetic labels across documents. We take entities as nodes, relation types as edges, and frequencies of relation triplet as weights. Then, we fuse the above two knowledge graphs and calculate a consistency score of each relation triplet by its frequency in the two knowledge graphs, which can be formulated as follows:

$$s_{ijk} = F_{ijk}^s + F_{ijk}^p \quad (5)$$

where, $F_{ijk}^s, F_{ijk}^p$ is the frequency of relation triplet $(e_i, e_j, r_k)$ for knowledge graphs $KG_s$ and $KG_p$. By further considering wrong relational facts that might be introduced in the fused knowledge graph, we prune the fused knowledge graph $KG_f$ by consistency scores of relation triplets.

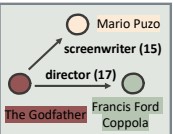 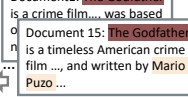

**Figure 5: An example of the knowledge expressed by different generated documents. The relation triplets (The Godfather, Francis Ford Coppola, director) and (The Godfather, Mario Puzo, screenwriter) are multiply expressed in different synthetic documents.**

Since frequencies of relation types are varied, we construct a dynamic threshold $\eta_k$ for each unseen relation $r_k$ to filter unreliable triplets, formulated as follows:

$$\eta_k = \overline{s_{ijk}} - \sqrt{\frac{1}{N_k^\eta - 1} \sum_{l=1}^{N_k^\eta} (s_{ijk} - \overline{s_{ijk}})^2}, \qquad (6)$$

where $s_{ijk}$ is the consistency score of the relation triplet $(e_i, e_j, r_k)$. $\overline{s_{ijk}} = \frac{1}{N_k^\eta} \sum_{l=1}^{N_k^\eta} s_{ijk}$ is the average of consistency scores of relation triplets with the relation type $r_k$. $N_k^\eta$ is the quantity of triplets that belong to unseen relation type $k$.

In our pruning strategy, we remove the relation triplet $(e_i, e_j, r_k)$ if its consistency score $s_{ijk}$ is lower than its threshold $\eta_k$. In this way, we can maintain useful knowledge and reduce the incorrect relational facts in the fused knowledge graph. We re-label the synthetic data with the denoised knowledge graph $KG_d$. Meanwhile, we also filter synthetic data that lacks valuable unseen relation triplets during the re-labeling process.

---

**Algorithm 1** GenRDK Training Procedure

**Define:** Seen data $DA_s$, triplets $T_s$, and relation type set $R_s$, Unseen data $DA_u$, triplets $T_u$, and relation type set $R_u$, Original synthetic data $DA_{syn}$ and triplets $T_{syn}$, Denoised synthetic data $\hat{DA}_{syn}$ and triplets $\hat{T}_{syn}$, Pseudo relation triplets: $T_p$, Knowledge graph: $KG$, Backbone model: $M$, Chain-of-retrieval prompt: $CoR$, Predict relation triplets: $TR$.

**Require:** $D_s, R, R_s, R_u$.
**Ensure:** $R_s \cap R_u = \emptyset$.
  1. $D_{syn} \leftarrow CoR\,(ChatGPT, R_u, R)$
  2. $\hat{M}_{pre-denoising} \leftarrow Train\,(M, DA_s, T_s, R_s)$
  3. $T_p \leftarrow Predict\,(\hat{M}, D_{syn}, T_{syn}, R_u)$
  4. $KG_s \leftarrow T_{syn}$
  5. $KG_p \leftarrow T_p$
  6. $KG_f \leftarrow Fusion(KG_s, KG_p)$
  7. $KG_d \leftarrow Prune(KG_f)$
  8. $\hat{DA}_{syn}, \hat{T}_{syn} \leftarrow Denoise\,(KG_d, D_{syn}, T_{syn})$
  9. $\tilde{M}_{ZeroDocRTE} \leftarrow Train\,(M, \hat{DA}_{syn}, \hat{T}_{syn}, R_u)$
  10. $TR \leftarrow Predict\,(\tilde{M}_{ZeroDocRTE}, DA_u, R_u)$
**return** $TR$

---

## 3.5 Relation Triplet Extractor

With the denoised synthetic data $\hat{D_{syn}}$, we train a relation triplet extractor by fine-tuning the generative language model LLaMA2-13B-Chat. The training process can be expressed as follows:

$$\tilde{M} \leftarrow Train(M, I, \hat{d}_i^{syn}, R_u, \hat{T}_i^{syn}), \qquad (7)$$

where $M$ denotes the backbone model. $I$ is the task description of the DocRTE task. $\hat{d}_i^{syn}$ is the $i$-th document in denoised synthetic datset $\hat{D_{syn}}$, $R_u$ is the unseen relation set, $\hat{T}_i^{syn}$ represents the denoised relation triplets of the $i$-th synthetic document, $\tilde{M}$ is the document-level relation triplet extraction model. We summarize the training procedure of the proposed framework GenRDKin Algorithm 1.

## 4 EXPERIMENTS

### 4.1 Datasets and Settings

We evaluate our framework on both zero-shot document-level relation and triplet extraction tasks with two public datasets. DocRED [31] is a popular large-scale human-annotated document-level relation extraction dataset with 96 pre-defined relation types, which is constructed from Wikipedia and Wikidata. Re-DocRED [26] is a revised version of DocRED by supplementing positive instances that are ignored in the DocRED dataset. We follow the previous zero-shot setting [2] that partitions the pre-defined relation types into a seen relation set and an unseen relation set. Only documents with labels of the seen set are available for training while documents that contain the unseen set are used for evaluation. The unseen relations are randomly selected from the relation types in datasets. For a fair comparison, we evaluate models under different sizes $m \in \{5, 10\}$ of unseen relation sets and randomly sample three times for each size to obtain different unseen relation sets.

The synthetic data generated by our proposed GenRDK can be used for both zero-shot document-level relation and triplet extraction tasks as we generate the whole document, entities, and triplets. Therefore, to illustrate the effectiveness of our framework, we conduct extensive experiments on both zero-shot document-level relation and triplet extraction tasks.

**Relation Triplet Extraction.** We adopt LLaMA2-13B-Chat as the backbone model. We use LoRA [11] which is a popular parameter-efficient fine-tuning method to fine-tune the LLaMA2-13B-Chat. We set the learning rate to 1e-6. The batch size is 20. The experiments are conducted on four NVIDIA RTX A6000-48G GPUs.

**Relation Extraction.** We adopt the graph-based DocRE model [23] as the backbone model, and apply RoBERTa$_{large}$ [16] as the context encoder. We use AdamW [17] as the optimizer. We set the learning rate to 3e-5. We apply warmup for the initial 6% steps. The batch size is 8 for both the training and test process. The experiments are conducted on a single NVIDIA RTX A6000-48G GPU. For both ZeroDocRTE and ZeroDocRE tasks, we use the $F_1$ as the evaluation metric to evaluate the performance of our framework on unseen relation types.

### 4.2 Baseline Methods

As zero-shot document-level relation and triplet extraction tasks are new task settings, we evaluate the performance of several popular LLMs on the above two task settings as benchmarks. Baseline

**Table 1: Experimental results on two public datasets for Zero-shot Document-level Relation Triplet Extraction (ZeroDocRTE). The CoR means the model trained by original synthetic data without our consistency-guided denoising strategy.**

| Model | Re-DocRED | | | | DocRED | | | |
|---|---|---|---|---|---|---|---|---|
| | m=5 | | m=10 | | m=5 | | m=10 | |
| | Dev | Test | Dev | Test | Dev | Test | Dev | Test |
| LLaMA2-7B | 2.4 ± 1.9 | 2.7 ± 1.9 | 1.2 ± 0.9 | 1.3 ± 0.8 | 2.3 ± 1.6 | 2.9 ± 2.3 | 1.2 ± 1.0 | 1.4 ± 1.0 |
| LLaMA2-7B-Chat | 4.9 ± 2.4 | 5.0 ± 3.0 | 3.6 ± 1.6 | 3.8 ± 1.8 | 4.8 ± 3.0 | 5.0 ± 3.2 | 4.3 ± 2.0 | 4.6 ± 2.3 |
| Flan-T5-XXL | 5.3 ± 1.4 | 4.8 ± 1.9 | 4.1 ± 1.4 | 3.7 ± 1.0 | 5.4 ± 2.3 | 6.0 ± 1.7 | 4.2 ± 1.1 | 4.5 ± 1.6 |
| LLaMA2-13B | 7.2 ± 2.3 | 7.1 ± 2.6 | 3.5 ± 0.6 | 3.1 ± 3.1 | 8.3 ± 2.2 | 8.1 ± 2.3 | 3.6 ± 0.6 | 3.8 ± 0.9 |
| LLaMA2-13B-Chat | 8.1 ± 1.6 | 8.7 ± 3.0 | 5.0 ± 1.0 | 5.2 ± 0.8 | 9.4 ± 2.0 | 9.0 ± 1.8 | 5.6 ± 1.1 | 5.5 ± 0.8 |
| ChatGPT | 11.2 ± 4.4 | 11.8 ± 3.8 | 7.5 ± 0.9 | 8.1 ± 1.5 | 14.7 ± 8.1 | 11.2 ± 5.1 | 8.5 ± 1.9 | 8.9 ± 2.3 |
| **Our Methods** | | | | | | | | |
| **CoR** | 11.0 ± 0.7 | 11.4 ± 2.3 | 6.6 ± 0.8 | 6.6 ± 1.1 | 13.1 ± 0.9 | 12.1 ± 1.0 | 7.6 ± 1.4 | 7.1 ± 0.6 |
| **GenRDK** | **13.3 ± 1.2** | **13.1 ± 2.6** | **8.2 ± 1.5** | **8.2 ± 0.6** | **15.2 ± 0.7** | **14.2 ± 1.3** | **9.2 ± 1.4** | **9.4 ± 0.6** |

methods includes LLaMA2-7B, LLaMA2-13B, LLaMA2-7B-Chat, LLaMA2-13B-Chat, Flan-T5-XXL, and ChatGPT. **Llama2** is an open-source LLM released by Meta, which is pretrained on publicly available online data sources. **Llama2-Chat** is the fine-tuned model that leverages reinforcement learning with human feedback. We evaluate the 7B and 13B versions for **Llama2** and **Llama2-Chat**. **Flan-T5** [4] is an encoder-decoder LLM released by Google, which is pre-trained on more than 1,800 language tasks. We evaluate the popular Flan-T5 XXL as the benchmark. **ChatGPT** is a powerful large language model based on reinforcement learning with human feedback released by OpenAI, which shows great ability in various NLP tasks.

## 4.3 Experimental Results

We compare our GenRDK framework with the above baselines for both ZeroDocRTE and ZeroDocRE tasks. The experimental results illustrate that our framework achieves significant performance improvement over the competitive baselines on two public datasets.

**Relation Triplet Extraction** As shown in Table 1, our framework GenRDK outperforms the previous baselines on both RE-DocRED and DocRED datasets. Specifically, when there are 5 different unseen relation types, our GenRDK achieves 13.1 ± 2.6 $F_1$ and 14.2 ± 1.3 $F_1$ on the test set of RE-DocRED and DocRED datasets, respectively. When the number of unseen relation types increases to 10, our GenRDK achieves 8.2 ± 0.6 $F_1$ and 9.4 ± 0.6 $F_1$ on the test set of RE-DocRED and DocRED datasets, respectively. We can observe that our model trained by original synthetic data outperforms the baseline model LLaMA2-13B-Chat by around 2.7 $F_1$ and 3.1 $F_1$ on the test set of RE-DocRED and DocRED datasets when $m = 5$. This suggests that our chain-of-retrieval prompt can effectively generate documents that contain unseen relational facts. Moreover, the performance of the model trained on denoised synthetic data improves by around 1.7 $F_1$ and 2.1 $F_1$ on the test set of RE-DocRED and DocRED datasets. This suggests the effectiveness of our consistency-guided cross-document knowledge denoising strategy.

**Relation Extraction** Our chain-of-retrieval prompt enables LLMs to generate the whole document, entities, and relation triplets.

| Vanilla Prompt | Chain-of-Though Prompt |
|---|---|
| Generate one fictional wikipedia style paragraph that contains at least 6 sentences and describes the relation type: <Unseen Relation Type>. Extract the possible relation triplets with the following relation types: <All Relation Types>. **Outputs:** #1. Title. #2. Generated paragraph. #3. Relational facts in JSON List format with following keys: head entity, tail entity, relation type, head entity type, tail entity type. | Perform the following instructions step by step: **Step one:** Generate one fictional wikipedia style paragraph that contains at least 6 sentences and describes the relation type <Unseen Relation Type>. **Step two:** Extract the entities in your above generated document. The entity type can be one of the following types: "Organization", "Location", "Time", "Person", "Miscellaneous", "Number", "Blank". **Step three:** Extract the possible relation types between the entity pair that exists one or more relation types of the following relation types: <All Relation Types>: **Outputs:** #1. Title. #2. Generated paragraph. #3. Relational facts in JSON List format with following keys: head entity, tail entity, relation type, head entity type, tail entity type. |

<Unseen Relation Type>: point in time, league, educated at, platform, child.
<All Relation Types>: head of government, country, place of birth, place of death, father, mother, spouse, country of citizenship, continent, instance of, …

**Figure 6: Illustration of vanilla and chain-of-thought prompt. Our chain-of-retrieval prompt can be seen in Figure 4. We generate different groups of data by the above prompts.**

Therefore, to further illustrate the effectiveness of our framework, we perform extensive experiments on document-level zero-shot relation extraction tasks. As shown in table 2, our GenRDK achieves 41.3 ± 8.9 $F_1$ and 41.5 ± 8.7 $F_1$ on the test set of RE-DocRED and DocRED datasets, when there are 5 unseen relation types. When the number of unseen relation types is 10, our GenRDK achieves 30.1 ± 4.2 $F_1$ and 31.4 ± 4.6 $F_1$ on the test set of RE-DocRED and DocRED datasets. Our GenRDK significantly outperforms the strong baseline ChatGPT by 19.6 $F_1$ and 17.9 $F_1$ on the test set of RE-DocRED and DocRED datasets. This demonstrates that our GenRDK can retrieve the implicit knowledge from ChatGPT. Moreover, the DocRE model trained on our denoised synthetic data outperforms the model trained on the original data by 4.2 $F_1$ and 3.0 $F_1$ on the test set of RE-DocRED and DocRED datasets when $m = 5$. This suggests that our knowledge denoise strategy can reduce the wrong relational facts by the consistency of LLMs. In addition, we can observe that the performance of ZeroDocRE is higher than ZeroDocRTE. This is because the ZeroDocRTE task needs to extract the entity pair

Table 2: We present experimental results for Zero-shot Document-level Relation Extraction (ZeroDocRE) on two public datasets: RE-DocRED and DocRED. The CoR is trained by original synthetic data without our consistency-guided denoising strategy. The GenRDK is trained by our denoised synthetic data.

| Model | Re-DocRED | | | | DocRED | | | |
|---|---|---|---|---|---|---|---|---|
| | m=5 | | m=10 | | m=5 | | m=10 | |
| | Dev | Test | Dev | Test | Dev | Test | Dev | Test |
| Flan-T5-XXL | 4.5 ± 2.2 | 3.1 ± 2.4 | 1.6 ± 0.6 | 1.8 ± 0.9 | 4.0 ± 2.5 | 3.9 ± 2.3 | 2.1 ± 0.8 | 1.9 ± 0.7 |
| LLaMA2-7B-Chat | 4.9 ± 2.0 | 4.8 ± 1.7 | 3.1 ± 1.4 | 3.0 ± 1.0 | 5.8 ± 3.2 | 4.5 ± 2.1 | 3.7 ± 1.9 | 3.6 ± 2.1 |
| LLaMA2-13B-Chat | 12.2 ± 2.0 | 12.8 ± 2.1 | 8.7 ± 0.9 | 8.5 ± 0.9 | 12.5 ± 2.2 | 12.8 ± 2.3 | 9.5 ± 0.6 | 9.6 ± 0.5 |
| ChatGPT | 20.6 ± 7.2 | 21.7 ± 7.5 | 13.7 ± 2.3 | 13.0 ± 1.8 | 21.9 ± 3.6 | 23.6 ± 3.1 | 15.5 ± 0.9 | 15.4 ± 2.9 |
| **Our Methods** | | | | | | | | |
| **CoR** | 38.0 ± 9.7 | 37.1 ± 9.2 | 28.7 ± 4.2 | 28.0 ± 3.7 | 38.4 ± 10.6 | 38.5 ± 9.1 | 32.6 ± 3.7 | 31.5 ± 3.8 |
| **GenRDK** | **39.9 ± 10.9** | **41.3 ± 8.9** | **30.6 ± 3.6** | **30.1 ± 4.2** | **42.5 ± 10.6** | **41.5 ± 8.7** | **33.7 ± 4.0** | **31.4 ± 4.6** |

Table 3: Experimental results of models trained by different synthetic data generated by vanilla chain-of-thought and our proposed chain-of-retrieval prompt.

| Model | Re-DocRED | | DocRED | |
|---|---|---|---|---|
| | Dev | Test | Dev | Test |
| **+ZeroDocRTE** | | | | |
| Vanilla Prompt | 8.35 | 9.04 | 10.32 | 9.77 |
| Chain-of-Thought | 9.80 | 10.43 | 12.80 | 12.85 |
| Chain-of-Retrieval | **11.19** | **13.23** | **14.19** | **13.38** |
| **+ZeroDocRE** | | | | |
| Vanilla Prompt | 38.58 | 42.45 | 35.38 | 34.98 |
| Chain-of-Thought | 45.10 | 47.80 | 45.27 | 43.72 |
| Chain-of-Retrieval | **48.51** | **49.21** | **51.08** | **48.30** |

and relationship at the same time, which is much more challenging than the ZeroDocRE task.

## 5 ANALYSIS AND DISCUSSION

In this section, we conduct extensive experiments to further analyze the effectiveness of our proposed chain-of-retrieval prompt and consistency-guided knowledge denoising strategy. We also present the case study of denoising synthetic data. Furthermore, we perform an ablation study to analyze the individual contributions of each component in our framework.

### 5.1 Effectiveness of Chain-of-Retrieval

To demonstrate the effectiveness of our proposed chain-of-retrieval prompt, we leverage different prompts to generate labeled data with the same unseen relation types. We compare our proposed chain-of-retrieval prompt with the vanilla prompt and chain-of-thought prompt, which can be seen in Figure 6. As shown in Table 4, the DocRE model trained on the synthetic data generated by our chain-of-retrieval prompt achieves 49.21 $F_1$ and 48.30 $F_1$ on the test set of Re-DocRED and DocRED datasets. For the ZeroDocRTE task, the model trained on the synthetic data generated by our chain-of-retrieval prompt achieves 13.23 $F_1$ and 13.38 $F_1$ on the test set of Re-DocRED and DocRED datasets. We can observe that the

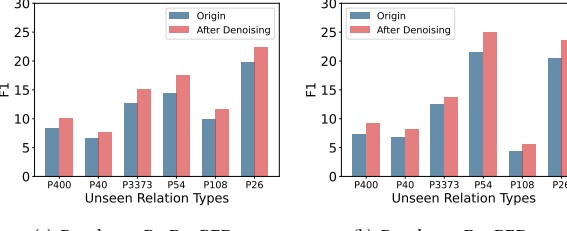

(a) Results on Re-DocRED.     (b) Results on DocRED.

Figure 7: Experiment results for different unseen relation types on Re-DocRED and DocRED datasets.

models trained on the synthetic data generated by our chain-of-retrieval prompt obtained significant performance improvements for both ZeroDocRTE and ZeroDocRE tasks. This demonstrates that our chain-of-retrieval prompt can effectively guide ChatGPT to synthesize document-level relation samples step by step.

### 5.2 Effectiveness of Knowledge Denoising

To intuitively demonstrate the effectiveness of our consistency-guided cross-document knowledge denoising strategy. We present extensive experimental results of different popular DocRE backbone models [30, 38] trained on original and denoised synthetic data. As shown in Table 4, it can be observed that all backbone models obtain performance improvement after training on the denoised synthetic data. To intuitively demonstrate the denoising effects, we present the performance of several unseen relation types. As shown in Figure 7, we can observe that the performance of different unseen relation types significantly improves with the denoised synthetic data on both Re-DocRED and DocRED datasets. This suggests that our denoising strategy can improve the quality of generated synthetic data.

### 5.3 Case Study

We present several examples of synthetic data that have been denoised using our consistency-guided cross-document knowledge denoising strategy in Figure 8. It can be observed that our GenRDK is able to reduce label noises in synthetic data by 1) Adding correct relational facts by the cross-document knowledge graph, such as

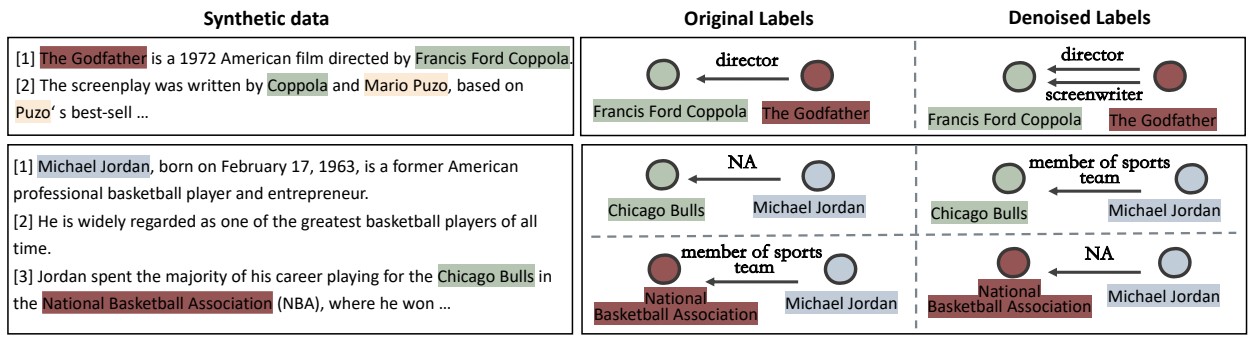

**Figure 8: Case Study. We present several samples with original and denoised labels of synthetic data.**

**Table 4: Experimental results of different DocRE backbone models trained on original and denoised synthetic data.**

| Model | Re-DocRED | | DocRED | |
|---|---|---|---|---|
| | Dev | Test | Dev | Test |
| **ATLOP-Bert-base** | | | | |
| +Synthetic Data | 45.73 | 45.48 | 46.11 | 45.30 |
| +Denoised Synthetic Data | 47.40 | 49.03 | 49.76 | 48.01 |
| **NCRL-Bert-base** | | | | |
| +Synthetic Data | 45.23 | 45.46 | 46.20 | 45.43 |
| +Denoised Synthetic Data | 47.37 | 46.37 | 48.01 | 47.69 |
| **Ours-Bert-base** | | | | |
| +Synthetic Data | 45.61 | 46.49 | 46.87 | 45.06 |
| +Denoised Synthetic Data | **48.02** | **49.19** | **49.97** | **48.05** |
| **ATLOP-Roberta-large** | | | | |
| +Synthetic Data | 48.43 | 48.74 | 48.38 | 47.75 |
| +Denoised Synthetic Data | 49.13 | 50.29 | 51.07 | 49.18 |
| **NCRL-Roberta-large** | | | | |
| +Synthetic Data | 46.73 | 48.00 | 46.09 | 46.19 |
| +Denoised Synthetic Data | 48.41 | 51.38 | 51.74 | 49.29 |
| **Ours-Roberta-large** | | | | |
| +Synthetic Data | 48.51 | 49.21 | 51.08 | 48.30 |
| +Denoised Synthetic Data | **50.61** | **51.88** | **52.90** | **51.31** |

the triplets *(The Godfather, Francis Ford Coppola, screenwriter)* and *(Michael Jordan, Chicago Bulls, member of sports team)*; 2) Reducing the false relational facts by the consistency of knowledge, such as the triplet *(Michael Jordan, National Basketball Association, member of sports team)*.

### 5.4 Ablation Study

To analyze the efficacy of each component within our GenRDK framework, we conduct an ablation study involving the removal of different components. As shown in Table 5, the performance diminishes with the removal of each component, showcasing the contribution of each component in our GenRDK framework. It can be observed that the removal of synthetic data leads to a 2.3 $F_1$ and 2.5 $F_1$ on the test set of Re-DocRED and DocRED. This drop demonstrates the effectiveness of synthetic data generated by our chain-of-retrieval prompt. When we remove our knowledge

**Table 5: Ablation study on the RE-DocRED and DocRED.**

| Model | Re-DocRED | | DocRED | |
|---|---|---|---|---|
| | Dev | Test | Dev | Test |
| **GenRDK** | **13.3 ± 1.2** | **13.1 ± 2.6** | **15.2 ± 0.7** | **14.2 ± 1.3** |
| w/o Denoising | 11.0 ± 0.7 | 11.4 ± 2.3 | 13.1 ± 0.9 | 12.1 ± 1.0 |
| w/o Seen Data | 12.2 ± 1.0 | 11.6 ± 0.5 | 12.7 ± 0.7 | 12.5 ± 0.9 |
| w/o Pruning | 11.9 ± 0.6 | 11.2 ± 1.6 | 13.0 ± 1.4 | 12.1 ± 0.6 |
| w/o Synthetic Data | 10.9 ± 1.0 | 10.8 ± 3.0 | 12.2 ± 0.8 | 11.7 ± 1.3 |

denoising strategy, the DocRTE trained merely by the original synthetic data achieves 11.4 ± 2.3 $F_1$ and 12.1 ± 1.0 $F_1$ on the test set of Re-DocRED and DocRED. This indicates that leveraging the consistency constraint of knowledge can improve the quality of synthetic data. We conducted a more fine-grained analysis of our knowledge denoising module. Experimental results illustrate that removing the pruning strategy or data with seen relation types results in varying degrees of performance degradation in the DocRTE model.

## 6 CONCLUSION

In this paper, we propose a novel document-level data generation and denoising framework for the challenging Zero-shot Document-level Relation Triplet Extraction task (ZeroDocRTE). Different from previous DocRTE models that heavily rely on human-annotated training data, our framework can distill the latent relational facts from LLMs and generate labeled data with new types of relations. To address the challenge of generating long-text data and multiple relation triplets, we propose a Chain-of-Retrieval prompt to guide ChatGPT to generate the document, entities, relation triplets, reasons, and support sentences step by step. To alleviate the inevitable noise in synthetic data, we construct cross-document knowledge graphs and propose a consistency-guided knowledge denoising strategy. To improve the quality of synthetic data, we remove unreliable relational facts by evaluating the consistency of knowledge. Leveraging the denoised synthetic data, we fine-tune the LLaMA2-13B-Chat for extracting document-level relation triplets. Experimental results demonstrate that our GenRDK outperforms competitive baselines on both DocRTE and DocRE tasks with zero-shot setting. In addition, extensive experiments illustrate the effectiveness of our denoising strategy. There are various challenges worth exploring, one potential avenue is enhancing the diversity and control of the generated data for ZeroDocRTE.

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
