# OpenReview forum: "Consistency Guided Knowledge Retrieval and Denoising in LLMs for Zero-shot Document-level Relation Triplet Extraction"
_ACM.org/TheWebConf/2024/Conference — TheWebConf24 Oral_

### Official Review · Reviewer_UHVe · 2023-11-06

**Novelty:** 5
**Technical Quality:** 5

**Review:**

The paper proposes GenRDK which is a new method for zero-shot document-level relation triplet extraction (ZeroDocRTE). Unlike existing methods that address sentence-level zero-shot relation extraction (ZeroRE) and relation triplet extraction (ZeroRTE), GenRDK addresses a more challenging task where relational facts can be expressed in multiple sentences or a document in general. GenRDK generates document-level synthetic data for the unseen relations using a new chain-of-retrieval (CoR) prompts that guide Chat-GPT to generate documents with the intended relations step by step. Then, the quality of the generated synthetic data is improved by a new denoising strategy based on the scores computed from synthetic and pseudo triples.

Pros

S1. The authors proposed to generate synthetic data for unseen relations using a novel CoR retrieval prompts that guide ChatGPT to generate a better quality synthetic data than both vanilla prompts and chain-of-thought (CoT) prompts.

S2. The authors showed that the generated synthetic data can be noisy with either incorrect triplets as a result of ChatGPT hallucinations, or missing triplets. To improve the quality of the synthetic data, the authors proposed to denoise the synthetic data with the help of pseudo triplets that are obtained from feeding a llama model, that is trained using seen triplets, with the generated documents.

S3. The experimental results show the effectiveness of GenRDK for both the ZeroRE and ZeroRTE tasks. The authors reported important ablation experiments such as the effectiveness of CoR and the effectiveness of knowledge denoising.

Cons

W1. The proposed model generates additional synthetic data for the unseen relations, which means that the training of the model contains data for the unseen relations, so the unseen relations are not completely unseen. This scenario is not the usual zero-shot learning where there is no knowledge about what is unseen.

W2. There should be more discussions about the reasons that make adapting existing sentence-level methods inadequate for ZeroDocRTE.

W3. From describing the baseline methods, it is unclear whether the reported baselines are fine-tuned on the seen data or not. If not, then the comparison can be unfair where the proposed method has the advantage of being fine-tuned before it is tested. In addition, the proposed method should be compared to other methods that also use additional generated data (like RelationPrompt [1] even if it is originally proposed for sentence-level ZeroRE and ZeroRTE) for a fair comparison.

[1] Chia et al. RelationPrompt: Leveraging Prompts to Generate Synthetic Data for Zero-Shot Relation Triplet Extraction

I acknowledge that I have read the rebuttal(s).

**Questions:**

In this paper, the authors proposed a new method, called GenRDK, for ZeroDocRTE. To improve the zero-shot relation extraction results, the proposed method generates document-level synthetic data using CoR retrieval prompts that guide ChatGPT to generate documents with the intended relations step by step. The generated data is denoised using a consistency-guided cross-document knowledge denoising. There are some points that should be taken into consideration:

1. Knowing what is considered as unseen and using that in the training strategy is an additional piece of information that should not be used in the typical zero-shot learning setting. It is like we actually know the expected relations in the testing set. Do we expect the same results to hold for really unseen relations? Please comment on this aspect. In addition, what is the size of the synthetic data? How does it affect the relation extraction results? Is the quality of the graph obtained from the synthetic data (KGs) better or worse than the pseudo triplets graph (KGp)?

2. The paper mentions that existing methods concentrate on sentence-level ZeroRE and ZeroRTE tasks, where entities and relations are confined within a single sentence. I actually don’t understand why the existing methods cannot be adapted to the document-level ZeroRE and ZeroRTE tasks. The motivation of the paper would be stronger if there are either more explanations about what makes the existing approach inadequate for document-level relation extraction tasks, or some evaluation metrics that show a weak performance of the existing sentence-level models applied to documents. Please comment on this aspect.

3. It is not clear whether the reported baselines are fine-tuned on the seen data or not.  If it is not, then the proposed method can clearly benefit from the fine-tuning using the seen data compared to the baselines. At least, the results of fine-tuning llama models on the seen data should be reported (which corresponds to the pseudo triplets). In addition, the proposed method can clearly benefit from the generated data for the unseen relations compared to the baselines even without synthetic data denoising. Therefore, the proposed method should be compared to other methods that also use additional generated data (like RelationPrompt [1] even if it is originally proposed for sentence-level ZeroRE and ZeroRTE) to clearly understand what makes GenRDK better than the baselines (ideally it should be CoR and denoising and not fine-tuning and synthetic data). Please comment on this aspect.

**Reviewer Confidence:**

3: The reviewer is confident but not certain that the evaluation is correct

**Scope:**

4: The work is relevant to the Web and to the track, and is of broad interest to the community

---

### Official Review · Reviewer_cKHu · 2023-11-07

**Novelty:** 3
**Technical Quality:** 5

**Review:**

This work focuses on the zero-shot document-level relation (triplet) extraction tasks. The authors argue that conventional methods mainly focus on the sentence-level zero-shot RE/RTE tasks, and propose this more challenging setting. To solve this task, the authors design a chain-of-retrieval prompt to guide GPT models to generate labeled data for training. To reduce noises, the consistency of cross-document knowledge is adopted to purify generated instances. The proposed framework achieves good results compared to LLM-based models on both RE and RTE.

In conclusion, this work is a good example of adopting LLMs for more challenging NLP tasks. The proposed framework is clear and sound. The experiments and analyses are solid.

**Questions:**

There are some questions as follows:

1.	Is there any zero-shot document-level RE/RTE methods?

2.	It suggested that the authors could give more discussions on existing works of sentence-level zero-shot RE/RTE, some of which may be compared as strong baselines in experiments. The current baselines are all LLM models.

3.	The authors could give more insights on how to design the chain-of-retrieval prompt. How to evaluate the indispensability of each prompt in model designing?

4.	Comparing Table 2 with Table 1, I find that the proposed framework achieves much larger improvements on RE than RTE (though RTE is the harder task). The authors could give more explanation on this finding.

**Reviewer Confidence:**

3: The reviewer is confident but not certain that the evaluation is correct

**Scope:**

2: The connection to the Web is incidental, e.g., use of Web data or API

---

### Official Review · Reviewer_bUsJ · 2023-11-20

**Novelty:** 4
**Technical Quality:** 4

**Review:**

This paper addresses the challenge of Zero-shot Document-level Relation Triplet Extraction (ZeroDocRTE), a fundamental task in information systems that typically demands substantial amounts of fully labeled data. The proposed framework, named Consistency Guided Knowledge Retrieval and Denoising (GenRDK), introduces an innovative approach leveraging advanced Large Language Models (LLMs), including ChatGPT and LLaMA. The key contributions can be summarized as follows:

1. **ZeroDocRTE Framework - GenRDK:** The paper introduces a novel framework for Zero-shot Document-level Relation Triplet Extraction, moving away from the reliance on human-annotated data. GenRDK utilizes advanced LLMs, specifically ChatGPT, guided by a Chain-of-Retrieval prompt to generate labeled long-text data systematically.
2. **Experimental Validation:** Extensive experiments are conducted on two public datasets, evaluating the effectiveness of GenRDK in both DocRTE and DocRE tasks under zero-shot conditions. The results demonstrate the superiority of GenRDK over competitive baselines, highlighting its capability to distill latent relational facts from LLMs.
3. **Knowledge Consistency Denoising Strategy:** The paper contributes a consistency-guided knowledge denoising strategy, leveraging cross-document knowledge graphs to remove unreliable relational facts and improve the overall quality of the synthetic data.
4. **Fine-tuning with LLaMA2-13B-Chat:** Leveraging the denoised synthetic data, the authors perform fine-tuning on the powerful LLaMA2-13B-Chat model, showcasing its effectiveness in extracting document-level relation triplets.

**Pros:**

1. The innovative framework, titled Consistency Guided Knowledge Retrieval and Denoising (GenRDK), specifically designed for Zero-shot Document-level Relation Triplet Extraction (ZeroDocRTE), has been introduced. This framework exhibits commendable performance on the DocRTE and DocRE datasets, indicating its efficacy in handling complex relation extraction tasks.
2. The methodology employed for generating supervised data through ChatGPT is noteworthy. It incorporates a consistency-guided knowledge denoising strategy, which not only assures the quality of the data but also significantly reduces the resource-intensive nature of manual data annotation.
3. The dataset and related code have been made open-source, ensuring reproducibility.

**Cons:**

1. Concerns arise regarding the consistency-guided knowledge denoising strategy. This strategy relies on refining two Knowledge Graphs (KGs), created from data generated by ChatGPT and pseudo-labels from a Pre-denoising Model. Given that both KGs are inherently prone to noise and uncertainty, the effectiveness of this denoising process remains questionable. A detailed analysis, including the proportions of different error types encountered during the denoising phase, particularly the errors in removal and addition, would be beneficial for a comprehensive understanding of this strategy.
2. The sensitivity of different models to varying prompts is a critical aspect that appears to be underexplored in the study. The reliance on a singular prompt template raises concerns about the generalizability of the results. Expanding the experimental design to include a diverse range of prompt templates and presenting an average of the outcomes would provide a more robust evaluation of the framework's performance across different scenarios.

**Questions:**

See cons.

**Reviewer Confidence:**

3: The reviewer is confident but not certain that the evaluation is correct

**Scope:**

3: The work is somewhat relevant to the Web and to the track, and is of narrow interest to a sub-community

---

### Official Review · Reviewer_RLck · 2023-11-29

**Novelty:** 5
**Technical Quality:** 6

**Review:**

This paper presents a zero-shot document-level relation extraction framework. The new framework aims to address the label data scarcity issue by leveraging LLM. Specifically, it first uses LLM to auto-generate labeled long-text data, then applies a unique denoising strategy to clean/filter the generated synthetic data, and finally uses the refined data to fine-tune a final docRE model (mainly based on LLaMA2-13B-Chat). Authors demonstrate the effectiveness of this framework through experiments on two public datasets and show it can outperform many strong baselines.

Overall this paper is clearly written, easy to digest, and the presented method is reasonable. The pipeline of using LLM to first generate synthetic data, do some domain knowledge guided filtering, and finally train a model is somewhat standard, but the authors do inject the ad-hoc task knowledge into the design of chain-of-retrieval prompt. Therefore I think some high-level idea of this paper is worth learning.

My main concerns of this paper are: 1) how well it can generalize? and 2) limited baseline methods. Two evaluated benchmarks originate from the same source and seems no other specific zero-shot docRE models are tested (c.f. Table 1). BTW, I feel the experiments in Table 4 are more important and are not emphasized enough. Finally, although the authors claim they will release the code and synthetic dataset on GitHub later but they haven't done so during the review period.

To summarize, I feel the paper worth reading but there are a few places can be further improved.

**Questions:**

1. Have you tried to generate a non docRE based dataset and test your framework's effectiveness on the dataset?
2. Is  your framework sensitive to the (likely hand-picked) chain-of-retrieval prompt and query LLM?

**Reviewer Confidence:**

3: The reviewer is confident but not certain that the evaluation is correct

**Scope:**

4: The work is relevant to the Web and to the track, and is of broad interest to the community

---

### Decision · Program_Chairs · 2024-01-22

**Decision:**

Accept (Oral)

**Comment:**

Paper describes a zero-shot method to extract entities and semantic relationships from documents. The paper is clear and understandable, and the topic is a good fit with the conference. Overall, I think the approach is interesting and the experiments support the claims of the paper. The authors promise a code and data release after publication.

 The reviewing process has proceeded in an ideal fashion. Any questions raised by the reviewers are fully addressed by the authors, and in some cases the reviewers have signed off on these responses, expressing their support for acceptance.

 The topic has relatively broad interest, and I recommend oral presentation. However, I note that I am not well calibrated on the breakpoint between oral and poster presentation.